# Unveiling dynamic hepatocyte plasticity in HepaRG cells with a dual CYP reporter system

Riku Asano[1], Yohei Iizaka[2], Makoto Kashima[3], Yojiro Anzai[2], Shinpei Yamaguchi ®[1]*, Masako Tada[1]†

1 Stem Cells & Reprogramming Laboratory, Department of Biology, Faculty of Science, Toho University, Chiba, Japan, 2 Department of Microbiology, Faculty of Pharmaceutical Sciences, Toho University, Chiba, Japan, 3 Department of Biomolecular Science, Faculty of Science, Toho University, Chiba, Japan

† Deceased.
* shinpei.yamaguchi@sci.toho-u.ac.jp

**Data Availability Statement:** All RNA-seq datasets have been deposited in the NCBI Gene Expression Omnibus (GEO) with the accession number GSE272892.

## Abstract

Primary hepatocytes are widely utilized for investigating drug efficacy and toxicity, yet variations between batches and limited proliferation capacity present significant challenges. HepaRG cells are versatile cells, capable of maintaining an undifferentiated state and differentiating through dimethyl sulfoxide treatment, allowing for molecular analysis of hepatocyte plasticity. To elucidate the underlying molecular mechanisms of HepaRG cell plasticity, we used CYP3A4G/7R HepaRG cells engineered to express DsRed under the control of the fetus-specific CYP3A7 gene and EGFP under the adult-specific CYP3A4 gene promoter. In time-lapse imaging of CYP3A4G/7R HepaRG cells, we observed CYP3A7-DsRed expression transitioning from negative to positive during proliferation period and CYP3A4-GFP expression activating during differentiation. The de-differentiation potency of differentiated CYP3A4G/7R HepaRG cells was assessed using inhibitors and cytokines. It was found that Y-27632 (Y), A-83-01 (A), and CHIR99021 (C) (collectively referred to as YAC), which are known to promote liver regeneration in mice, did not induce CYP3A7-DsRed expression. Instead, these inhibitors increased CYP3A4-GFP expressing population. Furthermore, CHIR99021 alone increased CYP3A4-GFP-positive cells, while Wnt3a treatment increased CYP3A7-DsRed-positive cells, suggesting that Wnt signaling plays distinct roles in HepaRG cells. It was apparent that de-differentiated cells had increased CYP3A4 activity after a second round of differentiation, compared to differentiated cells after the first round. Transcriptomic analysis of HepaRG cells revealed distinct profiles between proliferative, differentiated, and de-differentiated states, highlighting their robust plasticity. Notably, hepatoblastic cells de-differentiated by YAC or C displayed transcriptome patterns similar to undifferentiated cells, whereas CYP3A7-DsRed and CYP3A4-GFP exhibited expression patterns different from those of undifferentiated cells. These findings underscore the dynamic nature of HepaRG cells while cautioning against solely relying on CYP3 family gene expression as a marker of differentiation.

**Funding:** This work was supported by the Japan Society for the Promotion of Science (JSPS) [23H02397 to S.Y.], and the Toho University Grant for Research Initiative Program [TUGRIP 2022 to M.T. and S.Y., TUGRIP 2024 to S.Y.]. The funders had no role in study design, data collection and analysis, decision to publish, or preparation of the manuscript.

**Competing interests:** The authors have declared that no competing interests exist.

## Introduction

Drugs are absorbed in the small intestine and metabolized in the liver by cytochrome P450 isozymes (CYPs). About 50–60% of drugs metabolized by CYPs are oxidized by CYP3A4, a member of the CYP family [1]. In drug discovery and development, candidate compounds are evaluated for drug-drug interactions and hepatocyte toxicity using cells that express CYP3A4, and in many cases, primary cultures of human liver cells are used. The liver exhibits high regenerative potential, but transplantation is necessary when liver damage exceeds regenerative potential. The resection of the liver, resulting in the loss of 70% of the liver, has been shown to stimulate regeneration by dividing two nuclei of hepatocytes into one nuclear cell [2]. The liver stem and progenitor cells are oval cells and small hepatocytes, respectively. Small hepatocytes reside within the small bile duct and their periportal area. The isolated small hepatocytes can also differentiate into mature hepatocytes *in vitro* [3]. HepaRG cells and primary hepatocytes each present distinct advantages and limitations as liver models. Upon differentiation, HepaRG cells express CYP3A4, making them well-suited for high-throughput drug screening. In contrast, primary hepatocytes, particularly in 3D spheroid cultures, closely replicate the metabolic and toxicological responses of hepatocytes *in vivo* [4]. However, primary hepatocytes suffer from low reproducibility due to significant batch-to-batch variability in cell characteristics. Their limited proliferative capacity also makes them unsuitable for high-content screening and contributes to their high cost. Additionally, primary hepatocytes exhibit low expression levels of CYP family genes, with previous studies showing that CYP3A4 expression is gradually silenced within four days of culture [5]. Primary hepatocytes may be more reliable for predicting in vivo outcomes, but evaluating drug metabolism requires caution. Understanding the specific characteristics and plasticity of each cell model is crucial for selecting the appropriate system for a given research objective.

Recently, it was reported that mature hepatocytes can be de-differentiated into undifferentiated proliferative cells by a mixture of three compounds, a Rho-associated kinase inhibitor Y-27632, a TGF-β receptor inhibitor A-83-01, and a GSK-3 inhibitor CHIR99021, which were referred to as YAC [6, 7]. YAC can reprogram mature hepatocytes to chemically induced liver progenitors (CLiPs) with a high nucleus/cytoplasm ratio and differentiation potential. These cells successfully regenerated the liver of mice suffering from chronic injury without losing their ability to differentiate and proliferate. Meanwhile, human hepatocarcinoma-derived cell line HepaRG, which was established from the liver tumor portion of a hepatitis C donor, has drawn attention as an alternative to primary cultured human hepatocytes. HepaRG cells have CYP3A7-positive hepatoblast-like characteristics when they are proliferating, and can differentiate into CYP3A4-positive hepatocytes and bile duct epithelial-like cells if treated with dimethyl sulfoxide (DMSO) [8]. As a result of exposure to DMSO, HepaRG cells exhibit key hepatic characteristics, such as morphological changes, activation of drug-metabolizing enzymes, and drug-sensing nuclear receptors. In a low-cell density condition, differentiated hepatocyte-like HepaRG cells revert to hepatoblast status, but the molecular basis for plasticity has remained elusive [9, 10]. Particularly, little was known about the signaling pathways and compounds that regulate the differentiation and de-differentiation of HepaRG cells.

CYP3A4, which plays a critical role in drug metabolism, is regulated during development and its expression is suppressed until birth. Fetal hepatoblasts predominantly express CYP3A7, a liver-specific CYP3A family gene, rather than CYP3A4, which substitutes for its function. The expression of CYP3A7 in hepatocytes disappears after birth while that of CYP3A4 is induced around the central vein of the liver lobule in a region called zone 3 [11]. Previously, we generated dual reporter HepaRG cells containing GFP in CYP3A4 promoter and DsRed in CYP3A7 promoter to monitor CYP3A4 family gene expression in real-time (Fig 1A) [12]. DsRed-

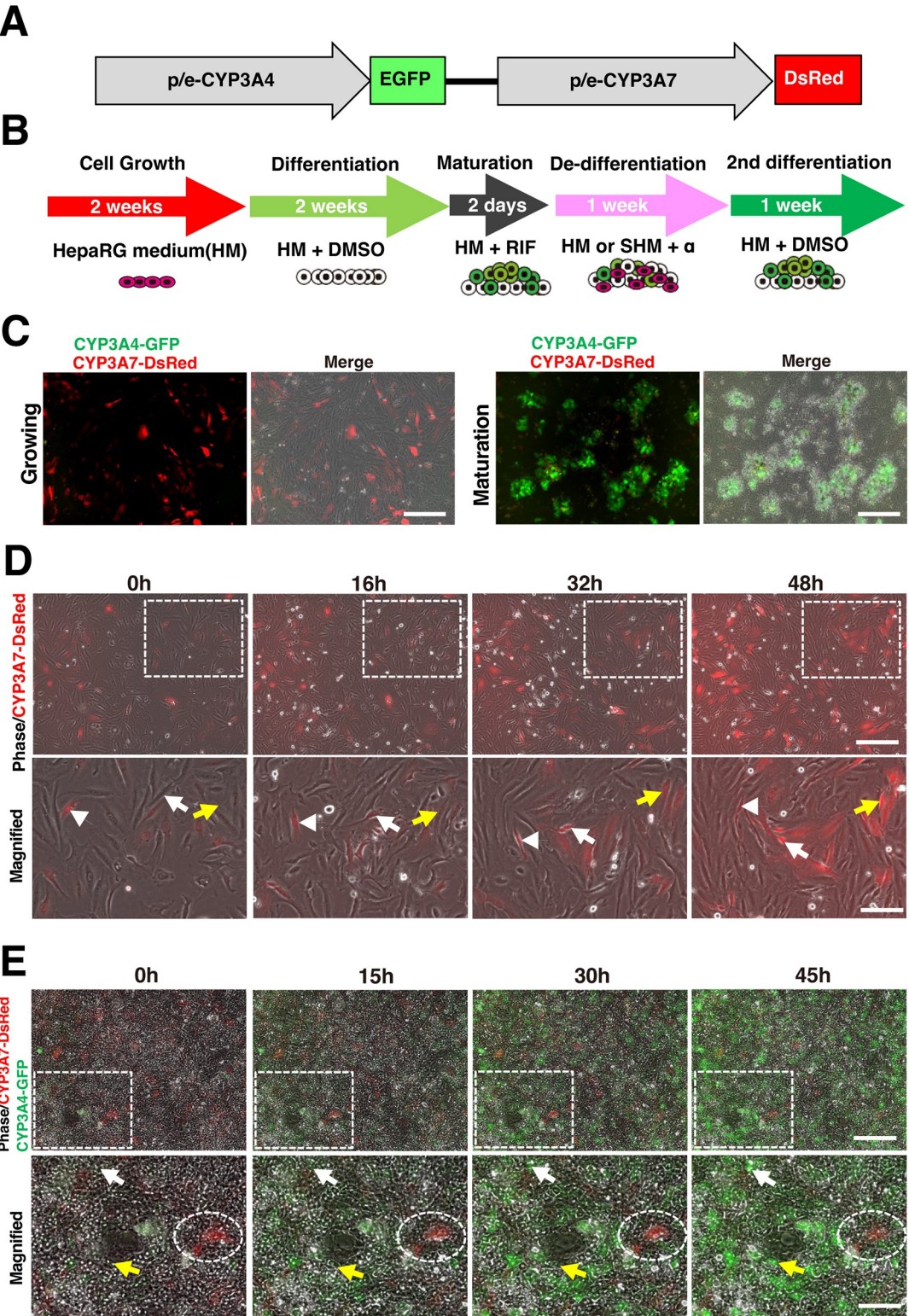

**Fig 1. Time-lapse imaging of the differentiation and maturation of CYP3A4G/7R HepaRG cells. (A)** Expression vector illustration for CYP3A4G/7R HepaRG cells. **(B)** Culture schedule, with arrows indicating the durations of each phase. An outline of the medium and reporter gene expression is provided below each arrow. **(C)** Representative CYP3A4G/7R HepaRG cells in the growing and maturation stages. Cells at day 10 of the growing phase (left) and day 2 of maturation (right) are shown. Scale bar, 100 μm. **(D)** Time-lapse imaging of CYP3A4G/7R HepaRG cells during the growing phase. A 48-hour

capture of cells from day 5 of the growth phase is shown. Dashed rectangles indicate areas in magnified images. Arrows and arrowheads indicate identical cells. Arrows point to cells that expressed CYP3A7-DsRed during the observation, while arrowheads indicate CYP3A7-DsRed-positive cells at the start of the observation. Scale bar, top 100 μm, bottom 50 μm. **(E)** Time-lapse imaging of CYP3A4G/7R HepaRG cells at maturation phase. A 45-hour capture of cells from day 1 of the maturation phase is shown. Dashed rectangles indicate areas in magnified images. Arrows and arrowheads indicate identical cells. Arrows point to cells that expressed CYP3A4-EGFP during the observation. The dashed ovals indicate the areas where CYP3A7-DsRed expression remained throughout the observation. Scale bar, top 100 μm, bottom 50 μm.

positive proliferating CYP3A4G/7R HepaRG cells disappeared after differentiation, and EGFP-positive cells gradually emerged. The fluorescence intensity of EGFP in differentiated cells was correlated with the activity of the CYP3A4 enzyme [12–14]. In this study, we utilized CYP3A4G/7R HepaRG cells to examine the effects of compounds on de-differentiation, as well as how de-differentiated cells could be redifferentiated. Furthermore, we demonstrated through transcriptome analysis that HepaRG cells are highly plastic, capable of switching between two distinct cell populations when differentiated and de-differentiated.

# Materials and methods

## Cell culture

In the process of passaging, cells were washed with PBS, treated with 0.05% Trypsin-EDTA (Gibco) for three minutes at 37˚C, then detach the cells using HepaRG medium (HM). We transferred the dissociated cells to a 15-mL tube, centrifuged at 200 G for 4 minutes, and resuspended the pellet in HM before seeding them into 24-well plates at a density of $6 \times 10^4$ cells/well. The differentiation process was initiated after 2 weeks of proliferation in 24-well plates with 2 days of 0.1% DMSO/HM followed by 1 day of 1% DMSO/HM and 11 days of 1.7% DMSO/HM. HepaRG differentiated cells were incubated with 10 m Rifampicin (RIF) for 2 days to induce maturation. As a basal media for de-differentiation, Small hepatocyte medium (SHM) was occasionally used: DMEM/F12 (Wako) containing 2.4 g/L $NaHCO_3$ and 1% L-glutamine, 30 mg/L l-proline (Nakalai tesque), 5% BSA (Sigma), 10 ng/ml epidermal growth factor (Sigma), insulin- transferrin-serine (ITS)-X (gibco), 10–7 M dexamethasone (Dex) (Sigma), 10 mM nicotinamide (Sigma), 1 mM ascorbic acid-2 phosphate (Wako), 1% sodium pyruvate solution (Nakalai tesque), and 50 μM hydrocortisone 21 hemisuccinate (Sigma) [15]. The de-differentiation was induced by 10M Y-27632 (Wako), 0.5M A-83-01 (Wako), 3M CHIR99021 (Miltenvi), or Y-27632, A-83-01, CHIR99021 in combination (YAC). For re-differentiation, cells were cultured in 0.2% DMSO/HM for two days, followed by 1% DMSO/HM for one day, and then 1.7% DMSO/HM for four days.

## Flowcytometry analysis

The cell suspension after dissociation with 0.05% Trypsin/EDTA was passed through a cell strainer cap tube (FALCON) and analyzed with CytoFLEX S (Beckman Coulter). P1 gates were set based on FSC and SSC to exclude dead cells. The measurement was continued until the number of cells within the P1 gate reached a total of 10,000. A 488 nm fluorescence detection channel for CYP3A4-GFP and a 561 nm fluorescence detection channel for CYP3A7-DsRed were used.

## CYP3A4 activity test

A 6-well plate of HepaRG cells was washed with PBS, then incubated for 60 minutes with 50 μM midazolam (Wako) in 1 mL of Minimum Essential Medium (Gibco). Afterwards, the supernatant was collected, mixed with 3 mL of methanol (Wako), and centrifuged at 200 G for

4 minutes. Five μL of the recovered supernatant was injected into a LC-MS/MS system. The LC-MS/MS analysis was performed on a LCMS-8040 (Shimadzu corporation, Kyoto, Japan) fitted with a STR ODS-II column (2.0 mm i.d. × 150 mm; Shinwa Chemical Industries, Tokyo, Japan) maintained at 35˚C. The mobile phase consisted of 0.1% formic acid in water (A) and acetonitrile (B) with the gradient elution performed at 0.2 mL/min using the following time program: from 0 to 2.0 min, linear gradient from 20 to 95% B; from 2.1 to 5.0 min, held at 95% B; from 5.1 to 7.0 min, liner gradient from 95 to 20%; from 7.1 to 15 min, held at 20% B for initialization. The MS was operated using electrospray ionization in positive ion-mode under the following operating conditions: heat-block temperature, 500˚C; desolvation line temperature, 300˚C; nebulizer gas flow rate, 3.0 L/min; drying gas flow rate, 15 L/min; collision-induced dissociation gas pressure, 230 kPa; ionspray voltage, 4.5 kV. Detection with the MS was performed using multiple reaction monitoring mode with 342.1 (Q1)/168.0 (Q3) for 1'-OH midazolam. 1'-OH midazolam in reaction mixture was quantified based on the linear calibration curves prepared by plotting the peak area ration derived from standard solution. A total protein level of the supernatant-harvested HepaRG cells was quantified in order to assess CYP3A4 activity per cell. Dissociated cells were lysed with 1% SDS/PBS containing the complete inhibitor (Merck). Sonication was performed with 30 cycles of 30 seconds on and 30 seconds off. A Pierce BCA Protein Assay kit (Thermo) was used to quantify the recovered proteins.

## RNA sequencing analysis

The HepaRG cells at the proliferation, differentiation, de-differentiation, and redifferentiation stages were harvested with 0.05% Trypsin-EDTA (Gibco) treatment. Total RNA of the resulting cell suspension was extracted using RNeasy Plus (QIAGEN). 3' RNA-Seq was conducted following the Lasy-Seq ver. 1.1 protocol (https://sites.google.com/view/lasy-seq/) [16, 17]. 50 ng of total RNA was reverse-transcribed using indexed primers and SuperScript IV reverse transcriptase (Thermo Fisher Scientific). Subsequently, all RT mixtures were pooled and purified using equimolar amounts of AMPure XP beads (Beckman Coulter, Brea, CA, United States) following the manufacturer's protocol. Double-stranded DNA synthesis was performed from the pooled samples using 5 U/μL RNaseH (Enzymatics) and 10 U/μL DNA polymerase I (Enzymatics). To avoid carryover of rRNA, RNase treatment with RNase T1 (Thermo Fisher Scientific) was performed, followed by purification of the samples using AMPure XP beads. DNA fragmentation, end repair, and A-tailing were carried out using 5× WGS Fragmentation Mix (Enzymatics). Ligation of Lasy-Seq adapters was performed using 5× Ligation Mix (Enzymatics, Beverly, MA, United States), followed by purification with AMPure XP beads. PCR cycles for library amplification were optimized by qPCR using EvaGreen, 20× in water (Biotium) and the AriaMx Real-Time PCR System (Agilent Technologies). Libraries were then amplified using KAPA HiFi HotStart ReadyMix (KAPA BIOSYSTEMS) on the ProFlex PCR System (Applied Biosystems) and purified with AMPure XP beads. Quality assessment of 1 μL of the library was performed by electrophoresis using the Bioanalyzer 2100 and Agilent High Sensitivity DNA kit (Agilent Technologies). Subsequently, 150 bp paired-end sequencing was conducted using the NovaSeq X Plus (Illumina). The reads were processed with fastp (version 0.21.0) using the following parameters: `-trim_poly_x -w 20 -adapter_sequence, AGATCGGAA GAGCACGTCTGAACTCCAGTCA`; `-adapter_sequence_r2: AGATCGGAAGAGCGTCGTGTAG GAAAGTGT`. Trimmed reads were then mapped to the human reference sequence Homo_sapiens.GRCh38.cdna.all.fa using BWA mem (version 0.7.17-r1188) [18] with default parameters. Read counts per gene were calculated with salmon using -l IU to specify library type (version v0.12.0) [19]. Subsequently, the total read counts per gene were calculated using R

(version 4.3.1). The following analysis was conducted Seurat (version 4.3.0) [20]. Briefly, read counts were normalized, scaled using the "NormalizeData" and "ScaleData" function with default parameters except for scale.factor = 10^6. After principal component analysis, UMAP dimensional reduction was conducted against principal components 1–15. Differentially expressed gene analysis was conducted FindAllMarkers with the following parameters: test.use = "DESeq2", logfc.threshold = 1. Gene Ontology enrichment analysis of biological process terms was conducted with clusterProfiler (version 4.8.2) [21, 22] and org.Hs.eg.db (version 3.17.0) [23] with FDR = 0.05.

## Results

### Switching of CYP3A gene expression during proliferation and differentiation

It has been reported that CYP3A4 gene expression switches from CYP3A7 to CYP3A4 during development, but its detailed kinetics remain unknown. A similar switch from CYP3A7 to CYP3A4 expression was observed in HepaRG cells during proliferation and differentiation. As HepaRG cells proliferated, CYP3A7-positive cells became dominant, but it was unclear whether CYP3A7-negative cells turned positive, or a small number of CYP3A7-positive cells actively proliferated. Hence, we examined reporter gene expression changes in growth cultures over two weeks. As previously reported, CYP3A7-DsRed-positive population in CYP3A4G/7R-HepaRG cells significantly increased after two weeks of proliferation (Fig 1B and 1C). A time-lapse imaging experiment from day 5 for two days showed that CYP3A7-DsRed-positive cells emerged from initially negative ones (Fig 1D). This change was mostly observed between 16 and 32 hours, corresponding to 0 to 6 hours of proliferation on day 6. These results suggested that positive cells arise from the negative cell population rather than preexisting CYP3A7-DsRed positive cells proliferating, which clearly demonstrated the plasticity of HepaRG cells. CYP3A4G/7R-HepaRG cells were differentiated after proliferation culture with high-concentration DMSO for two weeks, followed by maturation induction with 10 μM Rifampicin (RIF) for two days. Time-lapse imaging confirms CYP3A4-GFP-negative cells become positive within 15–30 hours in high-density sites (Fig 1E). The differentiated HepaRG cells exhibited a denser cytoplasm and a morphology similar to that of typical normal hepatocytes in primary culture, as well as effective activation of CYP3A4-EGFP, demonstrating the effectiveness of our culture method [24]. Meanwhile, CYP3A7-DsRed expression weakened during maturation, though some positive cells remained (Fig 1E).

### Evaluation of de-differentiation potential in CYP3A4G/7R HepaRG cells

We then examined mature differentiated HepaRG cell population to determine which de-differentiation conditions are most effective and to understand how plastic they are. A HepaRG medium (HM) and a Small Hepatocyte Medium (SHM) were used as the basic media for testing compounds Y-27632, A-83-01, CHIR99021, YAC, and Wnt3a. Wnt3a was included because it is known to enhance the proliferation of human fetal liver progenitor cells [25]. HM was widely used in HepaRG cell maintenance, while SHM, containing epidermal growth factor (EGF), was used to reprogram mature hepatocytes to undifferentiated state [6]. After one week of de-differentiation treatment, YAC did not yield significant CYP3A7-DsRed positive cells, while CYP3A4-GFP positive cells remained relatively unchanged (S1A and S1B Fig). Similar results were obtained with the CHIR99021 treatment. Contrary to this, treatment with Y-27632, A-83-01, or Wnt3a increased the number of positive CYP3A7-DsRed cells compared to pre-de-differentiation levels. However, neither of these compounds resulted in significant

changes in the ratio of CYP3A4-GFP positive cells (S1A and S1B Fig). De-differentiation treatment with Y-27632, A-83-01, or Wnt3a was found to activate CYP3A7-DsRed in initially negative cells, rather than converting CYP3A4-GFP positive cells into CYP3A7-DsRed positive cells, while CHIR99021 promotes self-renewal in CYP3A4-GFP-positive cells.

Flow cytometry analysis revealed a decrease in CYP3A7-DsRed positive cells from 47.5% to 9.1% and an increase in CYP3A4-GFP-positive cells from 0.4% to 19.5% during proliferation to maturation phases (Fig 2A and 2D). CYP3A7-DsRed-positive cells increased about 1.6-fold in the HM+Y-27632 and HM+Wnt3a-based de-differentiation conditions compared to post-maturation (9.1% to 14.3% in HM+Y-27632 and 14.3% in HM+Wnt3a), but only slightly in the A-83-01 and DMSO conditions. In contrast, a significant decrease was observed in CHIR99021(6.0%) and in YAC (5.0%). Following de-differentiation by SHM+Y-27632, SHM+A-83-01, and SHM+Wnt3a, CYP3A7-DsRed-positive cells increased 9.1% to 20.4% (2.2-fold), 17.6% (1.9-fold), and 18.7% (2.1-fold), respectively, while SHM+CHIR99021 and SHM+YAC decreased them (Fig 2B and 2C). It is suggested that Y-27632, A-83-01, Wnt3a, and SHM have an activating effect on CYP3A7 expression, while CHIR99021 suppresses it.

## Evaluation of differentiation potential of de-differentiated CYP3A4G/7R HepaRG Cells

A-83-01 or Wnt3a treatment did not affect the percentage of CYP3A4-GFP-positive cells, but increased CYP3A7-DsRed-positive cells, suggesting some degree of de-differentiation (Fig 2D). To confirm whether these cells had regained the ability to differentiate into hepatocytes, we performed second round of differentiation with a high concentration of DMSO in HM for one week (Fig 1B). All conditions tested demonstrated efficient differentiation as evidenced by the appearance of dense hepatocyte-like colonies (Fig 3A). Consistently, CYP3A7-DsRed-positive cells were decreased during differentiation, including HM+Y-2763 (from 15.0% to 8.9%), HM+A-83-01 (7.0%), and HM+Wnt3a (7.9%) (Fig 3A and 3B). However, these ratios were similar to HM+DMSO (8.8%), suggesting that simple prolonged culture might promote spontaneous de-differentiation and regained differentiation potential (Fig 3B). With SHM as a basic medium, CYP3A7-DsRed-positive cells decreased in SHM+Y-2763 (by 18.0% to 12.4%); SHM+A-83-01 (10.2%); SHM+Wnt3a (12.2%); and SHM+DMSO (10.6%). The percentage of CYP3A4-GFP-positive cells also decreased in SHM condition; from 16.0% to 11.1% in SHM+Y-2763, 12.0% in SHM+A-83-01, and 11.7% in SHM+DMSO. The ratios were lower than those of the first differentiation (Figs 2D and 3B). Based on the expression of the dual reporter, the results so far indicate that de-differentiation with tested conditions for one week was not sufficient to restore differentiation potential to the same levels as growing HepaRG cells.

## Evaluation of the metabolic activity of HepaRG cells post differentiation and redifferentiation

Although there was no significant increase in CYP3A4-GFP-positive cells after the second differentiation round, 11.1% to 18.8% of cells were found to be CYP3A4-GFP-positive (Fig 3B). In order to determine the metabolic activity of CYP3A4 in these cells, the catalytic activity of the oxidative reaction of midazolam to 1'-hydroxy (1'-OH) midazolam was measured [26].The cells at various stages including proliferation, maturation, de-differentiation, and the second round of differentiation were treated with 50 μM midazolam for 60 minutes, and 1'-OH midazolam in the supernatant was analyzed with a Liquid Chromatograph-tandem Mass Spectrometer (LC-MS/MS) (S2A Fig). As predicted from no CYP3A4-EGFP expression (Fig 1C and 1D), proliferating cells showed no detectable CYP3A4 activity, whereas mature, differentiated cells displayed 33.3 pmol/min/mg activity (Fig 3C and S2B Fig). It was commonly observed

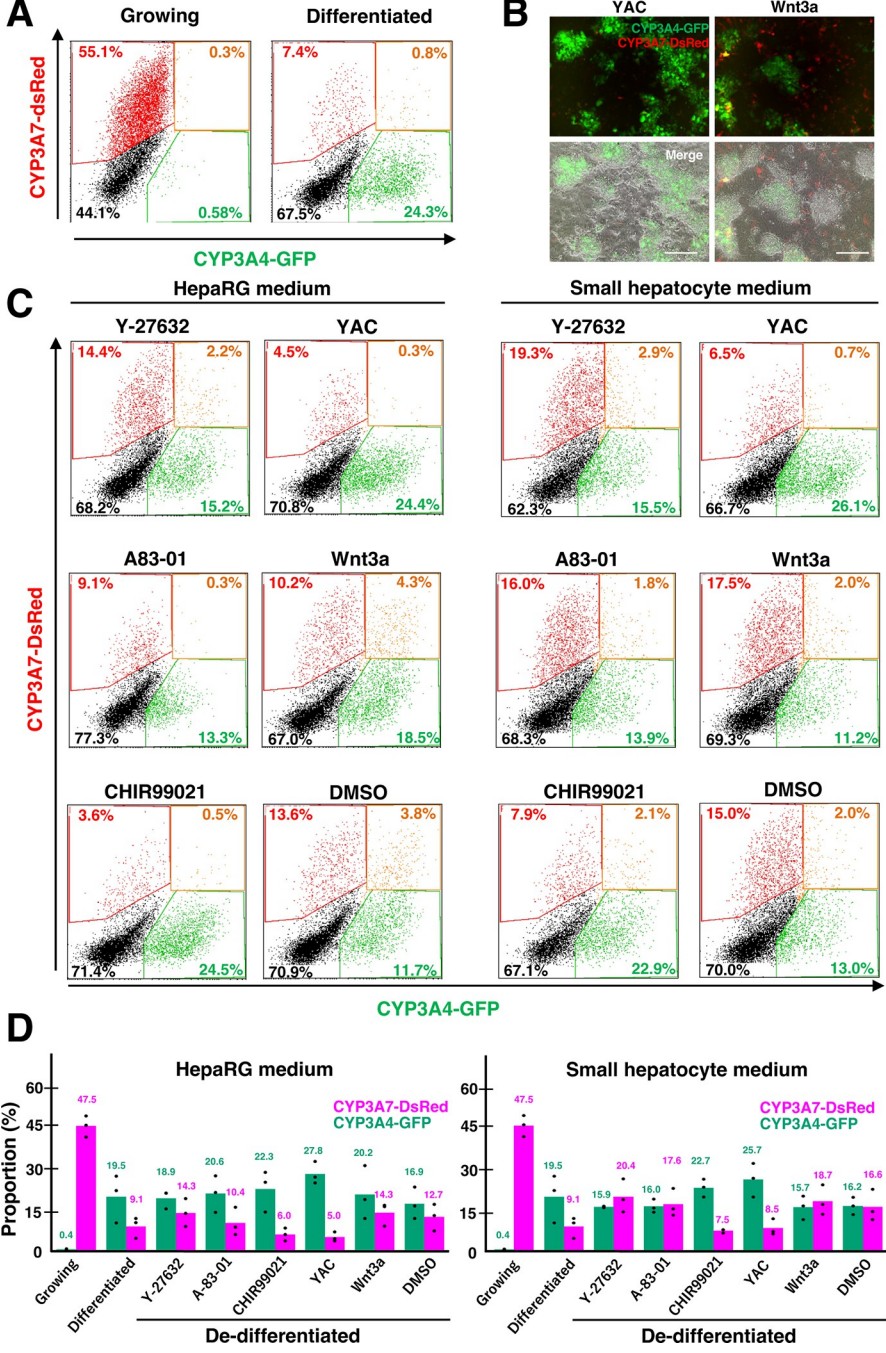

**Fig 2. A quantitative analysis of CYP3A4G/7R HepaRG differentiation and de-differentiation. (A)** Quantitative analysis of fluorescence expression in CYP3A4G/7R HepaRG cells during proliferation and after differentiation. Representative results of flow cytometry at day 10 of proliferation and day 2 of maturation were shown. **(B)** Representative images of CYP3A4G/7R HepaRG cells after 7 days of de-differentiation with YAC or Wnt3a in HepaRG medium. Scale bar, 100 μm. **(C)** Representative results of flowcytometer-based quantification of fluorescence expression in CYP3A4G/7R HepaRG cells after 7 days of de-differentiation with various compounds. **(D)** Summary of CYP3A4-GFP- and CYP3A7-DsRed-positive cells in CYP3A4G/7R HepaRG cells after 7 days of de-differentiation with various compounds. Actual values for each trial are shown as dots and averages as bars. n = 3.

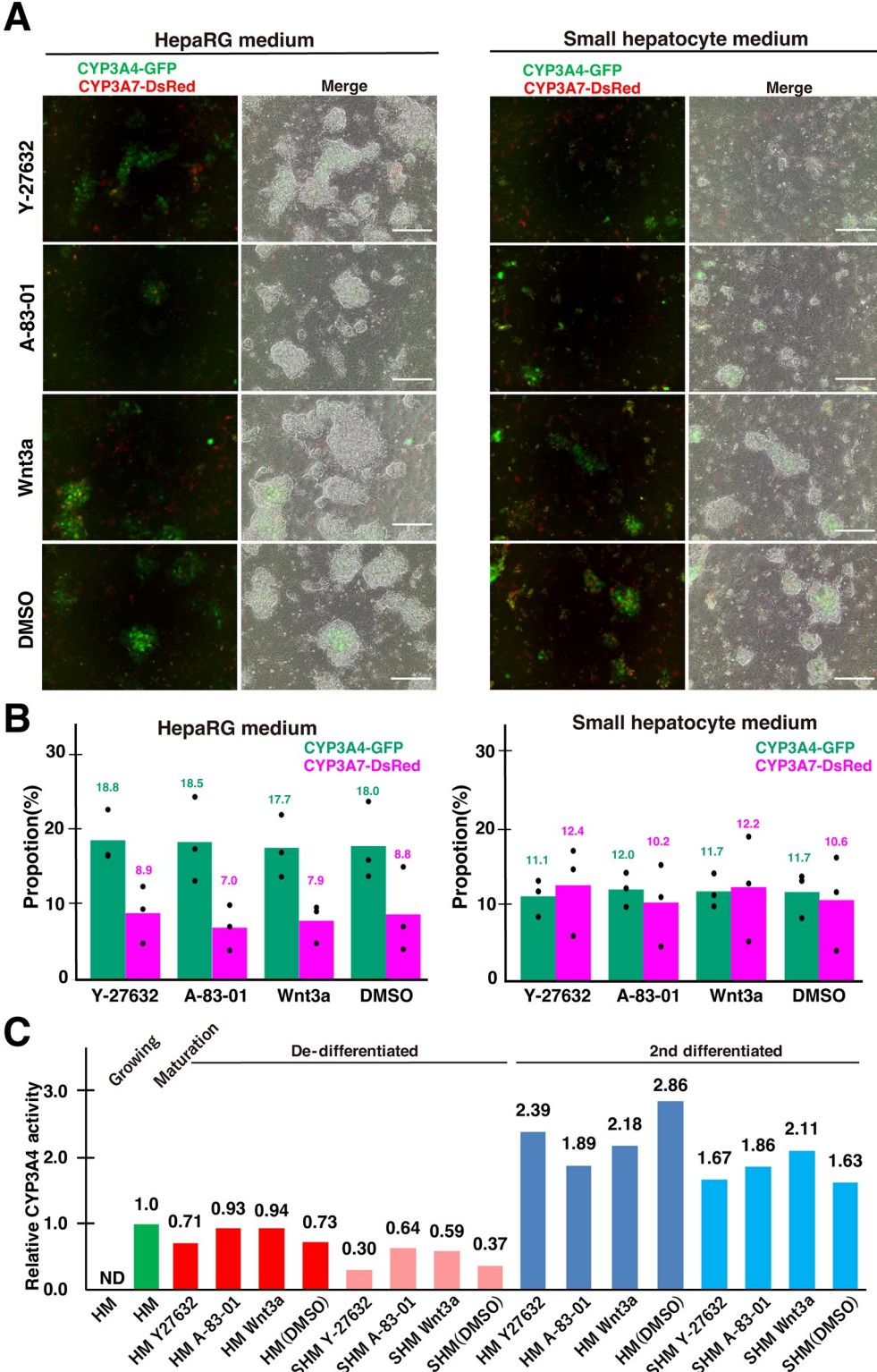

**Fig 3. Second round differentiation of de-differentiated CYP3A4G/7R HepaRG cells.** (**A**) Representative images of CYP3A4G/7R HepaRG cells on day 7 after redifferentiation after de-differentiation with the indicated compounds. Scale bar, 100 μm. (**B**) Summary of flowcytometry-based quantification of fluorescence expression in 7-day differentiated CYP3A4G/7R HepaRG cells that have been de-differentiated with the indicated compounds. Actual values for each trial are shown as dots and the averages as bars. n = 3. (**C**) The relative activity of CYP3A4 during

proliferation, maturation, and second differentiation. Individual enzymatic activities of CYP3A4 were quantified by detecting 1'-OH midazolam using LC-MS/MS. The concentration of 1'-OH midazolam was normalized by the total protein amount of the cells, and a relative value was calculated with the maturation cells set at 33.3 pmol/min/mg = 1.

that de-differentiated cells had reduced CYP3A4 activity by 7 to 70% compared to differentiated matured cells (Fig 3C). For example, HM+Y-2763, HM+A-83-01, HM+Wnt3a, and HM+DMSO de-differentiated cells showed 71%, 93%, 94%, and 73% CYP3A4 activity, respectively, compared to differentiated matured HepaRG cells. As shown in Fig 3C, the reductions were larger when SHM was used for de-differentiation. Although the number of CYP3A4-GFP-positive cells was not significantly changed after the second round of differentiation, CYP3A4 activity of them was consistently increased up to 2.86-fold compared to the first differentiation (Fig 3C and S2C Fig). This trend was commonly observed in both HM-based and SHM-based de-differentiated cells. These results indicated that CYP3A4 activity cannot be accurately estimated solely based on the number of CYP3A4-GFP-positive cells under all conditions.

## Transcriptome analysis of differentiation stages and drug treatment conditions

To elucidate the molecular entities underlying the differentiation status and plasticity of HepaRG cells, gene expression profiles were comprehensively analyzed using RNA sequencing (RNA-Seq) at different experimental phases, including proliferation, differentiation, de-differentiation, and re-differentiation (Fig 4A). A UMAP analysis of RNA-Seq data revealed distinct clusters of differentiated and undifferentiated cells (Fig 4B). A cluster of differentiated cells included human adult liver cells (HAL) as controls, suggesting that HepaRG cells transitioned into an adult liver-like transcriptional state once differentiated. According to reporter gene expression analysis, YAC and CHIR99021 de-differentiated cells did not fully convert into undifferentiated cells (Fig 3B), but they belong to the same group of undifferentiated cells as cells treated with Y-27632 and A-83-01. In the differentiated cells, 30 genes were upregulated and 252 genes were downregulated based on an analysis of genes that were significantly changed between the differentiated and undifferentiated cells (FC >2, FDR = 0.05) (S1 Table). A Gene Ontology enrichment analysis of the up-regulated genes in differentiated cells revealed that metabolic genes such as *AGXT*, *APOA5*, and *CYP2A7* were significantly enriched, suggesting that up-regulation of metabolic processes in differentiated-cell-enriched samples (Fig 4C and 4D and S2 Table). The AGXT enzyme is necessary for the detoxification of glyoxylate, a toxic metabolite produced during amino acid and carbohydrate metabolism [27]. APOA5 is a key regulator of plasma triglyceride levels. It functions by promoting the hydrolysis of triglycerides into fatty acids and glycerol, which can then be used for energy or stored in adipose tissue [28]. A CYP2A7 enzyme is responsible for drug metabolism and the synthesis of lipids such as cholesterol and steroids. CYP2A7 genetic variations can significantly affect the efficacy and toxicity of these drugs [29]. On the other hand, Gene Ontology enrichment analysis of the genes that are downregulated in differentiated cells shed light on the down-regulation of a group of genes that are alternative to collagen metabolism and translation, such as PCOLCE and RPL12 (Fig 4E and 4F).

## Discussion

Using CYP3A4G/7R HepaRG cells, we visualized real-time dynamics of CYP expression during differentiation. Before this study, there was the possibility that minor populations with

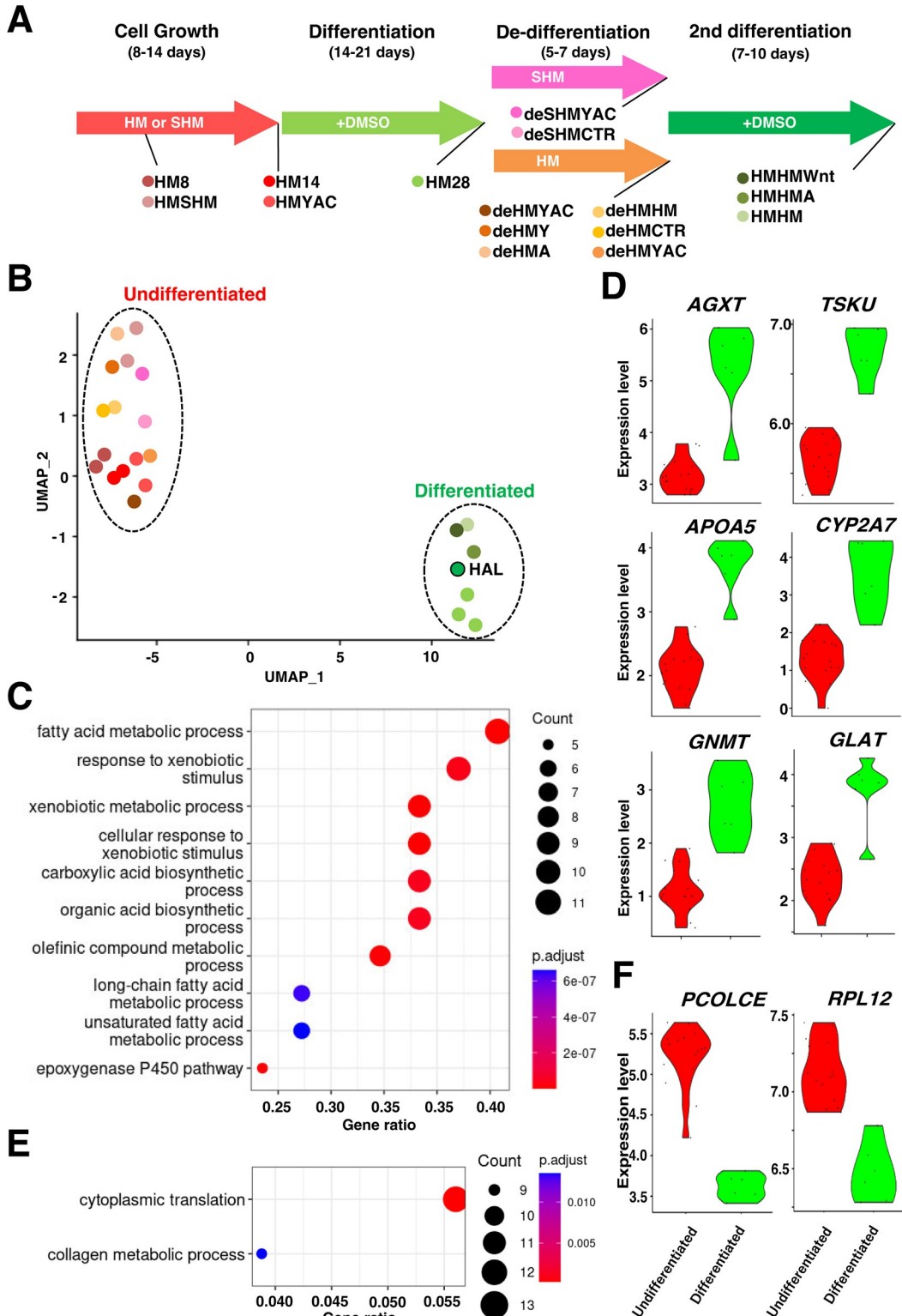

**Fig 4. Transcriptome of the differentiation stages of HepaRG cells.** (A) Overview of the culture schedule and RNA-Seq sample information. (B) UMAP representation of RNA-Seq results. The information about the dots is shown in (A). HAL: Human Adult Hepatocyte. (C) Gene Ontology analysis of up-regulated genes in differentiated HepaRG cells. (D) Violin plots of representative genes up-regulated in differentiated HepaRG cells. (E) Gene Ontology analysis of genes down-regulated in differentiated cells. (F) Violin plots of representative genes up-regulated in differentiated cells.

CYP3A7 expression might proliferate and become dominant. It was observed, however, that the CYP3A7-2DsRed-positive cells gradually emerged from CYP3A7-2DsRed-negative cells during proliferation, demonstrating that HepaRG cells are flexible enough to produce CYP3A7 expressing cells during proliferation (Fig 1D). Likewise, CYP3A4-GFP-positive cells emerge as cells mature and show increasing fluorescence intensity (Fig 1E). In densely populated areas, CYP3A4 expression was enhanced, while CYP3A7-DsRed expression was maintained, indicating that the cellular microenvironment and cell-cell interactions influence CYP3A gene expression.

The CHIR99021 treatment suppressed the increase of CYP3A7-positive cells during de-differentiation, whereas Wnt3a had only minor effects (Figs 3B and 5). In the previous study, CHIR99021-containing YAC treatment supported the proliferation of liver progenitor cells, and reprogramming of mature hepatocyte to CLiPs in both rodents and human [6, 7]. But, in this study, CYP3A7-positive population was not increased by YAC, rather CYP3A4-positive mature cells were increased. This could be due to the difference between HepaRG cells and primary culture hepatocyte. Both CHIR99021 and Wnt3a function in activating the Wnt signaling pathway. As a member of the Wnt protein family, Wnt3a is important for axis formation and neural crest cell development [30]. Wnt binds to receptors such as Frizzled, and regulates gene expression through two intercellular signaling pathways: the canonical pathway through β-catenin and the non-canonical pathway that is independent of β-catenin [31]. In the absence of Wnt signaling, β-catenin in the canonical pathway is degraded by GSK3β. However, when Wnt/Frizzled signaling is activated, this degradation is inhibited, leading to the stabilization of β-catenin. Once β-catenin enters the nucleus, it regulates the expression of target genes, such as Cyclin D1 and c-Myc. On the other hand, in the non-canonical pathway, Wnt bound to Frizzled induces the release of calcium ions and activation of calcium-dependent kinase II (CaMKII) [32]. CaMKII has been shown to induce glucose metabolism in hepatocytes and stimulate hepatic stellate cell proliferation and activation [31, 33]. The GSK3β inhibitor CHIR99021 plays a role in activating the canonical pathway, while Wnt3a can activate both canonical and non-canonical pathways. Differentiated hepatocytes may have the ability to

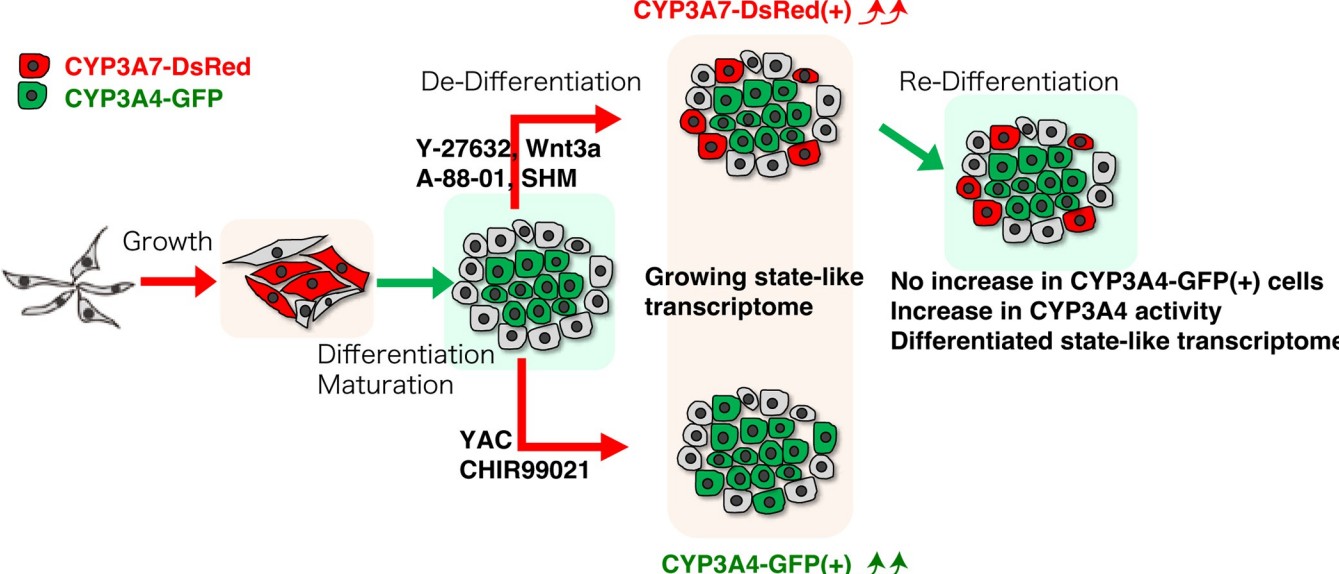

**Fig 5. An overview of the dynamics encompassing reporter gene expression, catalytic activity, and transcriptome profiles within CYP3A4G/7R HepaRG cells.**

proliferate via the canonical path, while they may de-differentiate via the non-canonical pathway. Further molecular analyses of both *in vivo* and *in vitro* approaches are required to clarify these points.

An LC-MS/MS analysis showed that the activity of the CYP3A4 enzyme was undetectable in the proliferative stage, but increased in the mature HepaRG cells (Fig 3C). However, CYP3A4 enzyme activity did not decrease as much as in proliferating cells after de-differentiation treatment and remained high. In line with this, the number of CYP3A4-GFP positive cells did not decrease after de-differentiation treatment (Fig 3B). It is possible that the 1-week de-differentiation treatment without passage did not sufficiently allow the cells to return to the undifferentiated state. However, de-differentiated cells showed similar transcriptional patterns to undifferentiated cells (Fig 4B). It is known that CYP3A4 protein is stable, and its half-life *in vivo* is over five days [34]. Thus, one week of treatment was likely enough to induce a de-differentiation of the transcriptional state of HepaRG cells, but GFP and CYP3A4 proteins remained.

On the other hand, after re-differentiation, CYP3A4 activity was significantly higher than after maturation, ranging from 1.6- to 2.8-fold (Fig 3C). After initial differentiation and after re-differentiation, the percentage of CYP3A4-GFP-positive cells was almost the same, which suggests that CYP3A4 activity may have increased in the CYP3A4-GFP-positive cells or may have even increased in CYP3A4-GFP-negative cells. Alternately, changes in cofactor expression and the intercellular environment may affect CYP3 activity. Accordingly, it is suggested that the 1-week de-differentiation treatment did not allow all cells to return to the undifferentiated state, and that the remaining differentiated cells matured further upon re-differentiation, leading to an increase in the activity of CYP3A4. This interpretation is supported by the findings that cells with more CYP3A4-GFP-positive cells after de-differentiation showed higher CYP3A4 activity after redifferentiation (Fig 4D). To achieve a more accurate induction, CYP3A4-GFP-positive and CYP3A4-GFP-negative cells will need to be separated and redifferentiated, although it is not technically challenging.

Differentiated and undifferentiated HepaRG cells exhibited clearly distinct expression profiles (Fig 4B). However, it is important to note that this analysis is profiling a heterogeneous population. Each population should thus be viewed as enriched with differentiated or undifferentiated cells. It is possible to obtain a more accurate molecular landscape by analyzing the transcriptome of a single cell at different stages of differentiation. Although there is technical limitation, we identified 282 differentially expressed genes between differentiated and undifferentiated HepaRG cells based on current bulk RNA-seq analysis. Gene expression analysis showed that differentiated cells exhibited an increased expression of several metabolic genes, including *AGTX* and *CYP2A7* (Fig 4D). Collectively, while some points also need to be considered, our CYP3A4G/7R HepaRG cells are a powerful tool for analyzing HepaRG cell plasticity.

## Conclusions

In this study, we utilized a dual CYP3A4G/7R reporter HepaRG cell model to investigate the dynamic plasticity of hepatocyte differentiation and de-differentiation processes. Our findings demonstrate the flexible nature of HepaRG cells, with distinct molecular responses to differentiation and de-differentiation stimuli. Notably, the inhibitors YAC and CHIR99021 promoted CYP3A4-GFP expression, while Wnt3a treatment selectively increased CYP3A7-DsRed-positive cells, highlighting divergent roles of Wnt signaling in HepaRG cell plasticity. Despite the strong induction of CYP3A4 activity following re-differentiation, CYP3A4 expression and activity remained stable even after partial de-differentiation, suggesting incomplete reprogramming to an undifferentiated state. Transcriptomic analyses further revealed the distinct

molecular signatures associated with differentiated, undifferentiated, and de-differentiated states, underscoring the heterogeneity of these populations. Future single-cell analyses will be necessary to gain a more granular understanding of the molecular landscape during HepaRG cell state transitions. Collectively, our study highlights the utility of the CYP3A4G/7R HepaRG model as a powerful tool for probing hepatocyte plasticity and underscores the need for careful consideration of CYP gene expression as markers for differentiation. These findings will lead to a refined protocol for liver regeneration and drug metabolism studies, with potential implications for studying liver disease models and drug development for hepatic disorders.

## Supporting information

**S1 Fig. De-differentiation of HepaRG cells. (A,B)** Representative images of de-differentiated cells with chemicals in HepaRG medium (A), and in Small hepatocyte medium (B).
(TIF)

**S2 Fig. CYP3A4 enzyme activity in differentiated HepaRG cells. (A)** A calibration curve for 1'-OH Midazolam using LC-MS/MS. **(B)** The LC-MS/MS peak of 1'-OH midazolam used to measure the CYP3A4 enzyme activity in HepaRG cells. On days 7 of proliferation and day 2 of maturation, midazolam, an enzyme target of CYP3A4, was administered to cells. **(C)** The LC-MS/MS peak of second round differentiated cells.
(TIF)

**S1 Table. The gene list of differentially expressed genes in differentiated and undifferentiated HepaRG cells.**
(XLSX)

**S2 Table. GO terms enriched in the differentially expressed genes in differentiated HepaRG cells.**
(XLSX)

## Acknowledgments

We sincerely thank Dr C. Guguen-Guillouzo, Dr A. Jamin, and Dr C. Chesne (Biopredic International, France) for expert advice on HepaRG cells We gratefully acknowledge the technical support of Ms. Akari Mine and Dr. Shota Okuyama. Furthermore, we would like to thank Dr. Chizuka Obara (Henmi) and Dr. Tetsuya Muramoto for providing us with their critical comments on the manuscript. Finally, we would like to acknowledge Dr. Masako Tada, who passed away before this paper was published, for her contribution to this paper and to science.

## Author Contributions

**Conceptualization:** Shinpei Yamaguchi, Masako Tada.

**Data curation:** Riku Asano, Yohei Iizaka, Makoto Kashima, Shinpei Yamaguchi, Masako Tada.

**Formal analysis:** Riku Asano, Makoto Kashima, Masako Tada.

**Funding acquisition:** Yojiro Anzai, Shinpei Yamaguchi, Masako Tada.

**Investigation:** Makoto Kashima, Shinpei Yamaguchi, Masako Tada.

**Methodology:** Riku Asano, Yohei Iizaka, Shinpei Yamaguchi.

**Project administration:** Shinpei Yamaguchi.

**Supervision:** Makoto Kashima, Shinpei Yamaguchi.

**Validation:** Shinpei Yamaguchi.

**Visualization:** Shinpei Yamaguchi.

**Writing – original draft:** Shinpei Yamaguchi.

**Writing – review & editing:** Riku Asano, Yohei Iizaka, Makoto Kashima, Yojiro Anzai, Shinpei Yamaguchi.

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
