## [Decision Letter · Decision Letter 0]

29 Aug 2024

PONE-D-24-31783Unveiling Dynamic Hepatocyte Plasticity in HepaRG Cells with a Dual CYP Reporter SystemPLOS ONE

Dear Dr. Yamaguchi,

Thank you for submitting your manuscript to PLOS ONE. After careful consideration, we feel that it has merit but does not fully meet PLOS ONE’s publication criteria as it currently stands. Therefore, we invite you to submit a revised version of the manuscript that addresses the few points raised during the review process.

We look forward to receiving your revised manuscript.

Kind regards,

Isabelle Chemin, PhD

Academic Editor

PLOS ONE

Journal Requirements: When submitting your revision, we need you to address these additional requirements. 1. Please ensure that your manuscript meets PLOS ONE's style requirements, including those for file naming. The PLOS ONE style templates can be found at https://journals.plos.org/plosone/s/file?id=wjVg/PLOSOne_formatting_sample_main_body.pdf and https://journals.plos.org/plosone/s/file?id=ba62/PLOSOne_formatting_sample_title_authors_affiliations.pdf 2. Thank you for stating the following financial disclosure: "This work was supported by the JSPS (23H02397 to S.Y.), and the Toho University Grant for Research Initiative Program (TUGRIP 2022 to M.T. and S.Y., TUGRIP 2024 to S.Y.)." Please state what role the funders took in the study.  If the funders had no role, please state: ""The funders had no role in study design, data collection and analysis, decision to publish, or preparation of the manuscript."" If this statement is not correct you must amend it as needed. Please include this amended Role of Funder statement in your cover letter; we will change the online submission form on your behalf. 3. Thank you for stating the following in the Acknowledgments Section of your manuscript: "We gratefully acknowledge the technical support of Ms. Akari Mine and Dr. Shota Okuyama. Furthermore, we would like to thank Dr. Tetsuya Muramoto for providing us with his critical  comments on the manuscript. This work was supported by the JSPS (23H02397 to S.Y.), and the Toho University Grant for Research Initiative Program (TUGRIP 2022 to M.T. and S.Y., TUGRIP 2024 to S.Y.). Finally, we would like to acknowledge Masako Tada, who passed away before this paper was published, for her contribution to this paper and to science." We note that you have provided funding information that is not currently declared in your Funding Statement. However, funding information should not appear in the Acknowledgments section or other areas of your manuscript. We will only publish funding information present in the Funding Statement section of the online submission form. Please remove any funding-related text from the manuscript and let us know how you would like to update your Funding Statement. Currently, your Funding Statement reads as follows: "This work was supported by the JSPS (23H02397 to S.Y.), and the Toho University Grant for Research Initiative Program (TUGRIP 2022 to M.T. and S.Y., TUGRIP 2024 to S.Y.)." Please include your amended statements within your cover letter; we will change the online submission form on your behalf. 4. When completing the data availability statement of the submission form, you indicated that you will make your data available on acceptance. We strongly recommend all authors decide on a data sharing plan before acceptance, as the process can be lengthy and hold up publication timelines. Please note that, though access restrictions are acceptable now, your entire data will need to be made freely accessible if your manuscript is accepted for publication. This policy applies to all data except where public deposition would breach compliance with the protocol approved by your research ethics board. If you are unable to adhere to our open data policy, please kindly revise your statement to explain your reasoning and we will seek the editor's input on an exemption. Please be assured that, once you have provided your new statement, the assessment of your exemption will not hold up the peer review process. 5. Please review your reference list to ensure that it is complete and correct. If you have cited papers that have been retracted, please include the rationale for doing so in the manuscript text, or remove these references and replace them with relevant current references. Any changes to the reference list should be mentioned in the rebuttal letter that accompanies your revised manuscript. If you need to cite a retracted article, indicate the article’s retracted status in the References list and also include a citation and full reference for the retraction notice.

**Additional Editor Comments:**

Please follow up the reviewer's comments to improve the paper that is of interest in the field.

Reviewers' comments:

Reviewer's Responses to Questions

**Comments to the Author**

1. Is the manuscript technically sound, and do the data support the conclusions?

Reviewer #1: Yes

2. Has the statistical analysis been performed appropriately and rigorously? 

Reviewer #1: Yes

3. Have the authors made all data underlying the findings in their manuscript fully available?

Reviewer #1: Yes

4. Is the manuscript presented in an intelligible fashion and written in standard English?

Reviewer #1: Yes

5. Review Comments to the Author

Reviewer #1: The submitted PONE-D-24-31783 research paper “Unveiling Dynamic Hepatocyte Plasticity in HepaRG Cells with a Dual CYP Reporter System” describes a series of experiments aimed to disclose molecular mechanisms underlying the plasticity of the HepaRG cells. HepaRG cell line is often used as a substitute for primary human hepatocytes in drug efficacy, inactivation and toxicity testing. However, there is a putative problem related to its reported ability to dedifferentiate and re-differentiate with drastic changes of the spectrum of the expressed cytochrome P450 isozymes. The authors used a smart approach based on the use of the previously developed and elsewhere described “CYP3A4G/7R HepaRG cells engineered to express DsRed under the control of the fetus-specific CYP3A7 gene and EGFP under the adult-specific CYP3A4 gene promoter”. The results confirm high plasticity of the HepRG cell line suggesting caution when using it for drug testing. In my opinion the work is well planned and accurately fulfilled. I have only one suggestion to the authors: to elaborate on the differences between the HepRG cell line, primary hepatocytes in culture and hepatocytes in vivo and on how their results canbe used to improve drug testing. Also, Figures 2 and 3 contain a common misprint in the phrase “small hepatcyte medium”.

6. PLOS authors have the option to publish the peer review history of their article (what does this mean?). If published, this will include your full peer review and any attached files.

Reviewer #1: No

---

## [Author Response · Author response to Decision Letter 0]

6 Sep 2024

We appreciate the valuable feedback provided by the reviewer and have carefully addressed all comments. Below is a summary of the major revisions made to the manuscript:

Reviewer #1:

Comment 1: "I have only one suggestion to the authors: to elaborate on the differences between the HepRG cell line, primary hepatocytes in culture, and hepatocytes in vivo and on how their results can be used to improve drug testing."

Response: We appreciate this valuable suggestion. In the revised manuscript, we have expanded the introduction section to compare the characteristics of HepaRG cells with primary hepatocytes in culture and hepatocytes in vivo (lines 69-81). 

Comment 2: "Figures 2 and 3 contain a common misprint in the phrase 'small hepatcyte medium.'"

Response: We apologize for this oversight. The typo has been corrected to “small hepatocyte medium” in both Figures 2 and 3.

Editorial Requests:

1. Funding Statement Update: We have removed the funding information from the Acknowledgments section.

2. Data Availability Statement: We will make the full dataset available upon acceptance. The release date has been updated to Oct 1, 2024. We confirm that the data will be shared in compliance with PLOS ONE’s data-sharing policy.

3. Manuscript Formatting: We have ensured that the revised manuscript adheres to PLOS ONE’s formatting guidelines. The appropriate file naming conventions have been used, and changes are tracked in the highlighted version of the manuscript. 

We are confident that these revisions address all the concerns raised, and we believe the revised manuscript now meets the criteria for publication in PLOS ONE. We thank you again for your time and consideration.

---

## [Editor Report · Decision Letter 1]

17 Sep 2024

Unveiling Dynamic Hepatocyte Plasticity in HepaRG Cells with a Dual CYP Reporter System

PONE-D-24-31783R1

Dear Dr. Yamaguch,

We’re pleased to inform you that your manuscript has been judged scientifically suitable for publication and will be formally accepted for publication once it meets all outstanding technical requirements.

Kind regards,

Isabelle Chemin, PhD

Academic Editor

PLOS ONE

Additional Editor Comments (optional):

The authors did answer in a convincing manner to the questions raised during the review process.
---

## [Editor Report · Acceptance letter]

31 Oct 2024

PONE-D-24-31783R1 

PLOS ONE

Dear Dr. Yamaguchi, 

I'm pleased to inform you that your manuscript has been deemed suitable for publication in PLOS ONE. Congratulations! Your manuscript is now being handed over to our production team.

Kind regards, 

on behalf of

Mrs Isabelle Chemin 

Academic Editor

PLOS ONE